# Wearable Sensors for the Monitoring of Maternal Health—A Systematic Review

**DOI:** 10.3390/s23052411

**Published:** 2023-02-22

**Authors:** Anika Alim, Masudul H. Imtiaz

**Affiliations:** Department of Electrical and Computer Engineering, Clarkson University, Potsdam, NY 13699, USA

**Keywords:** FECG, FHR, fetal movement, m-health, PPG, wearable sensors

## Abstract

Maternal health includes health during pregnancy and childbirth. Each stage during pregnancy should be a positive experience, ensuring that women and their babies reach their full potential in health and well-being. However, this cannot always be achieved. According to UNFPA (United Nations Population Fund), approximately 800 women die every day from avoidable causes related to pregnancy and childbirth, so it is important to monitor mother and fetal health throughout the pregnancy. Many wearable sensors and devices have been developed to monitor both fetal and the mother’s health and physical activities and reduce risk during pregnancy. Some wearables monitor fetal ECG or heart rate and movement, while others focus on the mother’s health and physical activities. This study presents a systematic review of these analyses. Twelve scientific articles were reviewed to address three research questions oriented to (1) sensors and method of data acquisition; (2) processing methods of the acquired data; and (3) detection of the activities or movements of the fetus or the mother. Based on these findings, we discuss how sensors can help effectively monitor maternal and fetal health during pregnancy. We have observed that most of the wearable sensors were used in a controlled environment. These sensors need more testing in free-living conditions and to be employed for continuous monitoring before being recommended for mass implementation.

## 1. Introduction

Worldwide, more than 200 million people experience pregnancy annually [1]. Around 140 million births occur annually. The proportion of births attended by skilled health personnel was less than 81% in 2019 [2]. Pregnancy is a life stage that involves rapid physiologic and behavioral changes [3]. During pregnancy, a woman’s lifestyle, behavior, and physical activities can affect her health and that of the fetus [4]. Hence, constant monitoring is often required if there is risk or complication arises. Pregnancy complications are a severe maternal and infant health risk, leading to adverse outcomes such as miscarriage, preterm birth, stillbirth, and low birth weight [5]. Around 295,000 women die worldwide due to childbirth complications each year [2]. There are 23.8 maternal deaths per 100,000 births in the United States and 462 deaths per 100,000 births in low-income countries [6]. The most common causes of maternal morbidity and mortality are hemorrhage, hypertensive disorders, infection, and sepsis [7]. Improving maternal health is key to saving the lives of more than half a million women who die each year as a result of complications from pregnancy and childbirth [8].

There is a clear relationship between lifestyle behaviors, risk factors, and pregnancy complications. Factors affecting maternal health can be external, environmental, risk-based, physical, and behavioral factors [9]. Careful monitoring of vital signs and physical activities is essential for ensuring the mother’s and fetus’s health and safety during pregnancy. This monitoring is often the first step in the early detection of pregnancy abnormalities and risks, providing an opportunity for prompt and effective intervention to prevent maternal and neonatal morbidity and mortality [7]. 

In the traditional medical system, patients make appointments with doctors if symptoms appear and follow their advice until the problem resolves; this is often expensive and time-consuming due to frequent and scheduled visits to the experts [4,10]. The current pattern of prenatal care includes 15 face-to-face visits with providers [11]. The content of these visits includes essential health services, risk assessments, patient education, and the building of trust between patient and provider. In this traditional system, women miss out on important opportunities to monitor and understand their health and the health of their newborns [11].

With the recent advances in communications and technology, many devices and sensors have been developed to remotely monitor health conditions and track health parameters in daily environments. Wearable sensors, m-health technologies, mobile apps, and other wireless devices such as smartwatches open new possibilities for monitoring behavioral and physiological phenomena [12,13,14]. These devices allow blood glucose level, blood pressure, heart rate, and other biometric data to be consistently measured. The real-time information is then transferred to the healthcare providers. These devices facilitate two-way communication between the doctor and the patient [15]. In general, identifying physical activity based on wearable sensors is not an easy job and there are many challenges involved. For example, sensor placement on the body affects the physical recognition rate, where a certain physical activity may become more recognizable with sensors worn on specific body parts than others [4]. Therefore, a clear understanding of sensor principles and proper selection of wearables is essential for effective monitoring.

Several recent research studies have focused on the use of application-specific wearables to monitor maternal health during pregnancy. This review is intended to provide a systematic evaluation of wearable sensors for monitoring maternal health in daily life. The focus is to summarize the effectiveness and limitations of the state-of-the-art methods and to find the scope for future research in this area.

The rest of the paper is organized as follows. First, the review methodology is presented in Section 2, along with the research questions (RQ) specification. Section 3 and Section 4 present a detailed exploration of the review outcome. Section 5 provides a discussion, and Section 6 concludes with potential future works.

## 2. Review Methodology

The goal of this review was to analyze the adequacy of current wearable approaches for monitoring maternal health and fetal conditions and to identify the research gaps in current sensing methodologies. Sensors can be defined as a machine, modules, subsystems, or devices whose purpose is to detect changes and events in the environment [16]. The current review considers sensors that can collect data, send notifications/alerts (Mobile apps, Smartwatch), ambient sensors (Motion sensors, video camera, microphone), and are those capable of being worn or attached to the body (Belts, suits, wireless and flexible ECG and EEG sensors) as wearable sensors.

The search procedure was conducted according to the Preferred Reporting Items for Systematic Review and Meta-Analyses (PRISMA) guidelines [17]. There are several studies that follow PRISMA guidelines for systematic review, such as Imtiaz et al. [14], Bougea et al. [18], and Chowdhury et al. [19]. For this review, we adopted their processes. This methodology used the following four processes: (1) identifying the RQs, (2) recognizing the article sources, (3) searching articles based on the RQs, and (4) analyzing the search outcome.

### 2.1. Identifying Research Questions

Three research questions (RQs) were chosen to guide this systematic review:

(1) RQ1. What different sensor technologies are employed in prenatal monitoring tasks, how were those sensors placed on body location, and what were the data acquisition processes?

(2) RQ2. How was the acquired data preprocessed regarding noise and artifact removal and prepared for the algorithm development?

(3) RQ3. What specific information was extracted from those sensor signals, how were they extracted, and how was the performance of the sensor evaluated?

### 2.2. Article Databases Searched

The primary sources for the relevant literature were Google Scholar, IEEE Xplore, MDPI, Science Direct, IOP Science, PubMed, and the ACM Digital library. Table 1 shows the publication date of the 12 selected articles and their total citations.

### 2.3. Search Terms

The following free-text search terms were used: ‘wearable sensors’, ‘maternal health’, ‘fetal movement’, ‘fetal heart rate’, and ‘fetal ECG’. The search results were strictly restricted to the English language. The selected full-text articles/references were further analyzed for this review. The selection was further narrowed by applying the eligibility criteria described in Table 2.

Initially, through the database search, a total of 79 publications were identified, and 46 were chosen for title and abstract screening. 67 articles failed to satisfy the eligibility and were excluded. Finally, 12 articles were selected for the full-text review. Figure 1 demonstrates the flow diagram of the systematic review strategy.

### 2.4. Analyzing Review Outcome

A total of 12 publications were found on monitoring fetal and maternal health (including fetal heart rate, fetal movement, maternal physical activities, and stress) using wearable sensors and detecting abnormalities in the early pregnancy stage. The methodologies can broadly be categorized as Seismocardiography (SCG), a technique of measuring the vibrations produced by the beating heart [29], Gyrocardiography (GCG), a non-invasive technique for assessing heart motions by using a sensor of angular motion–gyroscope–attached to the skin of the chest [30], FECG (Fetal Electrocardiogram), which is a biomedical signal that gives an electrical representation of FHR (Fetal Heart Rate) to obtain vital information about the condition of the fetus during pregnancy and labor from the recordings on the mother’s body surface [31], or MECG (Maternal Electrocardiogram), determined by R-R interval and QRS complex measurement of the ECG signal. Fetal position and activity can determine Fetal Movement (FM) and the mother’s heart rate can also indicate the mother’s stress level. Table 3 outlines the sensor-acquired signals for each study.

## 3. Sensing Methodologies and Data Acquisition

### 3.1. FECG/FHR

A fetal electrocardiogram (FECG) signal may provide potentially precise information about fetal condition during pregnancy and labor. FECG characteristics such as heart rate, waveform, and dynamic behavior are convenient in determining fetal life, development, maturity, and the existence of fetal distress or congenital heart disease [32]. The studies reported in [7,20,21,22,23,24] were focused on FECG/FHR monitoring.

In study [7], a four-layer flexible printed circuit board was fabricated for the mother’s chest and a two-layer board for the abdomen. The chest sensor (Figure 2) measured maternal HR, RR (the time elapsed between two successive R-waves of the QRS signal on the electrocardiogram [33]), and central temperature. The chest sensor included a bio-potential analog front end (AFE) (MAX30001; Maxim Integrated), a high-frequency three-axis Inertial Measurement Unit (IMU) (LSM6DSL; STMicroelectronics), and a clinical-grade thermometer (MAX30205; Maxim Integrated) [34,35]. It was placed under the suprasternal notch (Figure 2).

The abdominal sensor measures FHR and uterine contractions. This sensor is placed just below the umbilicus (Figure 2). Figure 2 demonstrates sensor placement and capture signals from the patient.

In another study [20], a prototype electrometer-based amplifier was developed using Electric Potential Sensing (EPS) technology [36] for non-invasive monitoring of fECG in utero from the surface of the maternal abdomen. The prototype was built using a custom ultra-high input impedance EPS sensor with an internal input bias current circuitry and guarding. It was designed using four dry electrodes. The data was acquired on a laptop computer. Data display and storage were controlled by a custom-designed graphical interface based on LabVIEW software [37], which included a peak detection algorithm for HR values determination [20]. Figure 3a shows the electrode placement for maternal and fetal ECG recording.

In another study [24], fetal heart rate (FHR) was collected using seismocardiogram (SCG) and Gyrocardiogram (GCG) recordings from abdominal inertial sensors. Three commercially available wearable sensor nodes (Shimmer 3 from Shimmer Sensing [38]) were attached to the abdominal wall by elastic straps. One sensor node was placed at the center of the upper abdominal wall, close to the reference fCTG ultrasound probe. The two remaining sensors were attached to the lower part of the abdominal wall at symmetric positions (Figure 3B) [24].

The Accelerometers (Kionix KXRB5-2042, Kionix, Inc. USA [39]) record the seismocardiogram (SCG) signal and the gyroscopes (Invensense MPU9150, Invensense, Inc. USA [40]) measure the gyrocardiogram signal (GCG) [24].

A fourth study [21] used an ultrasound Doppler heartbeat detector and a toco pressure sensor resembling a standard fetal monitoring system. It’s an end-to-end low-cost wireless and mobile fetal monitoring system that employs a body-worn fetal monitoring device augmented with wireless networking technology to enable a new area of care, allowing anytime/anywhere monitoring. This device can provide clinical expertise asynchronously and remotely [21].

In the study by [22], the monitoring system was primarily composed of a data acquisition module, data transmission module, signal storage module, and signal analysis platform. Electrodes were attached to the skin in a certain way (Figure 3C) in order to collect pregnant women’s Abdomen Electrocardiography (AECG) signals [22]. Three linearly independent ECG electrodes were used to construct a surface ECG vector map [41]. The electrode position was designed with three acquisition channels, a reference point, and a left leg drive. The reference electrode point was 5 cm below the center of the pregnant woman’s navel. Three acquisition electrodes form a triangle around the navel. The left leg drive electrode was on the right side of the participant. This configuration was chosen because it maximizes the SNR [22]. Figure 3C shows the electrode position on the patient.

In this article [23], a mobile wearable measuring system was designed using a microcontroller, sensors, and a shield, further enhancing its already existing capabilities. The board acts as a sink of all signals. The sensor, located on the pregnant belly, sends necessary ECG signals to the sink for further processing. The development board can either store local sensor data or send them to a web platform via the shield. The shield is responsible for transmitting, receiving, or harvesting data. In this work, the utilized commercial shield could send and receive data using Wi-Fi [23].

Table 4 gives information regarding FECG/FHR data collection in the mentioned papers.

### 3.2. Fetal Movement (FM)

Fetal movement is one of the most important clinically observed indicators of fetal activities (such as the position, duration, and relative force of FM). Study [25] used a multipoint IMU to detect FM signals from the abdomen of the pregnant woman. The FM signals were detected immediately by evaluating the signal energy and the signal interval was extracted as the basis of analysis. A triangular measurement shape (Figure 4) was extended from a circle as the center point. There was an IMU sensor at each of the three corners. The center circle included a related signal processing circuit, an IMU sensor, a rechargeable battery circuit, and a Secure Digital (SD) memory card. The device could closely adhere to the pregnant woman’s abdomen. The casing was made of a thermoplastic elastomer and the triangulation point for measurement used a flexible flat cable as a signal transmission line, thus enhancing the softness of the casing [25]. The device design is shown in Figure 4.

In another study [27], a passive and wearable device with two accelerometers was designed to sense subtle motion in the abdomen of pregnant women in order to replace maternal perception for out-of-hospital fetal movement monitoring. Figure 5 shows the device.

A wearable device, similar to a belt, with an INS (Independent Navigation sensor) sensor was designed and fabricated to monitor fetal movement [27]. A tri-axial accelerometer was used to measure fetal vibration in the maternal abdomen [42]. The microcontroller was used to transfer and store data on the micro-SD card, timestamp the data, and note the mother’s other activities, such as laughing, coughing, and hiccupping. The device was developed to ensure the mother’s comfort and its ability to be worn over a long period. When taking measurements, the device was placed around the mother’s waist. The sensor was embedded on a rubber pad in order to thoroughly contact the mother’s abdomen. When selecting material for the belt, its ability to absorb perspiration, its effect on skin, its flexibility, and color were considered [27].

Table 5 listed the number of participants and the FM recording time mentioned in the papers.

### 3.3. Maternal Health

Maternal health is the health of women during pregnancy, childbirth, and the postpartum period. Maternal health revolves around the health and wellness of pregnant women, particularly when they are pregnant, at the time they give birth, and during child-raising [43]. Improving maternal health is key to saving the lives of more than half a million women who die as a result of complications from pregnancy and childbirth each year [8].

#### 3.3.1. Daily Activities and Lifestyle

During pregnancy, physical activity can either create or prevent health issues depending on the different stages of maternity. Therefore, it is important to track different activities. Smartphones, mobile sensors, GPS, ambient sensors, and mobile apps can help to monitor activities. PAR (physical activity recognition) systems based on body-worn sensors provide better results than those based on either ambient or mobile phone sensors [4].

During the study [4], 61 subjects at various stages of maternity performed ten physical activities: walking up/down stairs, cooking, eating, hand exercise, laundry, lying down, walking, front bending, side bending, and standing. Data were collected by installing the wearable sensor module at the wrist position on either the left or right hand. Sensors at the wrist position are easily installed and managed using a smartwatch. According to their physical condition, the participants performed each activity for between 2 and 5 min. A single wearable sensor module consisting of an accelerometer, a gyroscope, and temperature sensors was installed at the wrist position [4].

Tracking or monitoring health parameters in daily life environments and adapting daily activities during pregnancy is important. Maternal adaptations during pregnancy lead to changes in lifestyle behaviors that may impact pregnancy complications. In the study [5], they focused on lifestyle behaviors (physical activity, sleep, stress, diet, and weight management) that can be tracked using state-of-the-art wearable technology.

#### 3.3.2. Stress Monitoring

There is a major concern about pregnancy-associated stress and anxiety, which are key risk factors for various pregnancy complications involving the health of the mother and fetus [44,45]. Maternal adaptations to decrease stress levels are important to facilitate a successful pregnancy. This algorithm was designed to adapt to changes in heart rate that occur during pregnancy [46,47]. To indicate the stress level, this study focused on measuring heart rate at rest [28].

For the first dataset, the participants wore a Garmin vívosmart-2 device over a period of 7 months. Heart rate, activity, and sleep information were gathered from the extracted data. The stress was measured when the subject was at rest. The second dataset was obtained from a Garmin vívosmart-3 device with a stress level classification. The heart rate, sleep, activity, and stress classification were extracted from the data; these data were used to test the performance of the algorithm in a supervised setting [28].

#### 3.3.3. Temperature and Oxygen Level

In study [7], another type of sensor (a limb sensor) was used. The limb sensor measured photoplethysmogram (PPG), a simple optical technique used to detect volumetric changes in blood in peripheral circulation [48], and skin temperature. The limb sensor wraps around the index finger of the participant to collect PPG and peripheral skin temperature. Figure 2 shows the acquired signal.

### 3.4. Survey Based Report

Study [3] focused on the perception of pregnant women and their providers at a rural health clinic regarding the use of wearable technology to monitor health and environmental exposures during pregnancy.

An anonymous 21-question electronic survey was administered to family medicine or obstetrics and gynecology providers at a rural health clinic, while a 21-question paper survey was made for pregnant women who came to the clinic for prenatal care. One hundred and three individuals responded to the survey as patients and 28 health care responded to the provider survey. The recruitment of pregnant patients and their medical providers took place at the Mountain Area Health Education Center (MAHEC), a rural health clinic that serves the entire 16-county region of Western North Carolina (WNC) [3]. Figure 6 shows the patient and providers response regarding wearable sensors.

Table 6 gives the list of sensors used in the above-mentioned studies and their descriptions.

## 4. Data Processing and Algorithm Development

In the comprehensive vital signs monitoring study [7], the data were filtered by a modified Pan–Tompkins algorithm and the R-peak, which indicates maternal HR, was detected. SpO2 was calculated by filtering the red and infrared channels, detecting peaks, and calculating the pulse amplitudes ratio. RR was recorded using composite chest wall movement data obtained from the x and y-axes of the accelerometer and the ECG signal. Continuous temperature measurements on the chest and limb sensors obtained from direct readings allow the fever curve to be monitored.

In the fetal heart rate (FHR) detection using seismocardiogram (SCG) and Gyrocardiogram (GCG) study [24], the readings of the respective axes were first band-pass filtered to focus on the desired frequency components. A zero-phase infinite impulse response (IIR) band-pass filter with cut-off frequencies of 0.8 Hz and 50 Hz was used to pre-filter the SCG waveforms. The observation from GCG was similar to SCG. Therefore, the information from all three sensors was merged to enhance the signal quality of SCG and GCG separately. The preprocessed SCG and GCG signals were converted by CWT with a Morse wavelet [24].

In this study of an electrometer-based amplifier prototype [59], developed using EPS technology for fECG monitoring, the sensor output voltage was fed to an analog filter with cut-off frequencies of 0.5 Hz and 100 Hz during the amplification stage. This customized version of the sensor was used to record maternal and fetal ECG signals. The low noise levels achieved within the sensor design avoided the use of post-processing stages and allowed visualization of the QRS complex in the raw fetal ECG trace, as shown in Figure 7 [20].

The analog output was fed to a commercial National Instruments data acquisition system. The data were acquired using a laptop computer. Display and storage of the data was controlled using a custom-designed graphical interface based on LabVIEW software, including an algorithm for peak detection to determine HR values [20].

In the study of the toco sensor, an instrumentation amplifier with a gain of 100 amplified the signal to the ADC input range. Further baseline subtraction and gain adjustment were implemented in the gateway software. A gateway was used for local data storage and visualization and to communicate with the mobile data network to transmit data to the server [21]. Both the wireless gateway and Bluetooth module emitted nonionizing radiation at frequencies ranging from 1 to 2.5 GHz. The FCC limit on the Specific Absorption Rate (SAR), a measure of the rate of energy absorption by the body when exposed to an RF field [60], for cellular telephones is 1.6 W/kg. Figure 8 represents various timings for data transmission in the system.

In study [22], the signal of each channel was divided into six non-overlapping segments (10 s for each segment) and the average of their corresponding SampEn values was returned as the result of the current channel. The signal quality was assessed by comparing the SampEn value in each channel with a constant threshold, that was set to 1.5 for AECG recordings. The average SampEn value was greater than 1.5 for the channels that were regarded as poor quality and subsequently excluded. SampEn less than 1.5 was considered as a good-quality signal and reserved. If less than two channels were of good quality, the two channels with the penultimate and the smallest SampEn values were reserved. The notch filter was applied to remove the power line interference in this work. The combination of the Butterworth filter and median filter was applied to remove baseline drift and impulsive artifacts. The power line interference, baseline drift, and impulsive artifacts of the AECG were mostly removed after the signal noise-canceling step [22].

Many different types of filters and algorithms were used for data processing in these 12 publications. Table 7 gives a description of the filters.

In the IMU sensor study [25], all the signals received by the IMU went through an Inter-Integrated Circuit, the hardware filter filtered out the 60 Hz noise, and the data were sent to the MCU. The signals from various channels were processed by a Kalman filter to reduce noise and then the position, duration, and Relative Force (RF) of the fetal movement were determined from the signal interval generated by the energy evaluation [25]. The process of the software is shown in Figure 9.

Three main features were extracted from the signal to analyze fetal movement [25]:FM duration calculationFM relative force calculationFM position evaluation

In the duo accelerometer study [26], the denoising process was applied. The model was set as the baseline of the data and subtracted from raw data to remove 0 Hz noise. The Wavelet denoising tool, ‘denoise,’ with the specific parameters ‘Wavelet’, ‘sym2’, ‘Denoising Method’, BlockJS’, ‘NoiseEstimate’, ‘LevelIndpendent’ in MATLAB, was applied. Another tool, ‘hampel’, was applied to detect and remove outliers [23,59].

After analyzing the fetal movement waveform, three main groups of features (statistical, morphological, and wavelet features) were selected. Minimum, maximum, standard deviation, mean, and median common statistical features were extracted. Absolute area, relative area, absolute area of the differential, entropy, and kurtosis were also extracted. The ‘Haar’ wavelet was used to decompose the data and extract the approximate coefficients of the discrete wavelet transform. The mean, median, and standard deviation of the wavelet transform were also selected as features for classification. Since each axis had 13 features, the total number of features was 78 [26].

Predictive models to map X to Y were built using a convolution neural network. The Convolutional Neural Network (CNN) architecture was adopted from [63].

In the PAR study [4], two different window sizes (one second and two seconds) and two configurations (overlapping and non-overlapping) were selected. In the one-second window size, the 0.5 s. Mean, SD (Standard Deviation), cosine similarity, RMS (Root Mean Square), Skewness, Kurtosis, Max value, Min value, Frequency Domain Features, Entropy, Zero Crossing, Quartile Range, and Absolute Time Difference between Peaks features were extracted from the data. A total of 43 features were used to represent the sensor data. K-Nearest Neighbors, Decision Tree, Random Forest, Induction rules, and Gradient boosted trees classifiers were used on the proposed MPAR system. Each classifier follows the general rule of supervised machine learning algorithms, where the classifier parameters are trained with the help of a training set and then its classification/recognition performance is evaluated with a completely disjoint test set. The dataset of this study is available upon request [4].

Table 8 depicts the sensors or devices that were used for monitoring maternal or fetal health. The number of participants in each study and the sensors’ communication technology is also shown.

## 5. Discussion

This study provides a systematic review of the existing wearable sensors for monitoring maternal health during pregnancy. This review analyzes all twelve available research articles describing wearable sensors and maternal and fetal health. It focuses on three research questions: different sensors module, preprocessing of the signal, and recognition or detection method. Table 7 and Table 8 give a summary of the employed sensors and data-obtaining methods.

For measuring FECG and FHR-ESP, Toco, prototype, or commercial sensors were used [7,20,21,22,23,24]. These sensors were placed on the abdominal wall and chest of the mother to measure MECG. Maternal heart rate was calculated to determine the stress level. IMU sensors and accelerometer sensors were used to measure fetal movement [25,26,27]. Limb sensors and wearable sensor modules on the wrist measured the mother’s body temperature and physical activities [4,7]. The data collected through these sensors were connected to smartphones and laptops, helping to display and store the data.

Data collection time was different for different methodologies. For studies [25,26,27], data collection for fetal movement detection was 1 h, 30 min, and 20 min respectively. For FECG and MECG recording, the participants had to remain in certain positions to ensure uninterrupted recording [21,22]. For [20,22,24], the recording was 30 s, 7 min, and 15 min respectively. The data collection was carried out during different stages of pregnancy, with no subject monitored throughout the whole pregnancy period.

Filtering is found to be required to remove external noise or artifacts. Most research studies used a notch filter to remove the power line noise [65] and an FIR or IIR filter to remove the artifact from the signal. Some devices have built-in filters and so they do not need to perform filtering externally. The study [23] developed a board employing a microcontroller, sensors, and a shield. This board performs the further processing of the FECG signal after acquiring the signals. In ref. [20], the study developed a prototype device that amplifies the fetal ECG signal and performs the analog filtering. The other studies primarily use commercially available sensors and perform the data processing and filtering in MATLAB, LabVIEW, Raspberry-PI, and other software. This section also covers the feature extraction methods. Most relevant features were extracted from the data before proceeding to the next step. Table 6 gives an idea about the extracted features in these studies.

These studies used machine learning and deep learning methods to evaluate or detect the signal. Table 9 shows the pre-processing methods and performance of the research analysis.

From Table 8, we can see that only 3 of the studies achieved an accuracy above 90% [22,25,28]. After reviewing the articles, we can propose that further work and development are needed in the field of wearable sensors for monitoring the health of pregnant women. All the wearable sensors mentioned in the articles used in monitoring were tested in a lab environment or clinical trial. Furthermore, the monitoring processes carried out in the papers reviewed here were conducted for a limited time and only during a certain gestational period. The number of participants is also limited and more rigorous testing is required [4,7,20,21,22,23,24,25,26,27,28]. Only in study [7] were bandage sensors implemented; these are more flexible than other sensors. Some pregnant women expressed discomfort wearing these sensors [25]. Maternal physical activity recognition was conducted for only ten activities [4]. For continuous monitoring during daily activities throughout the whole pregnancy, there will be many motion artifacts in the acquired signal. These are hard to recognize and remove. Table 10 describes the pros and cons of the sensor modalities used in the studies.

The sensors or devices used in these studies are safe for both mother and their child and most are commercially available. None of the studies claim any risk in using their sensors for monitoring fetal or mother health. The experiments were supervised by professional doctors [21,23,24,25]. The subject experimental protocol of studies [3,4,22,26,27,28] were approved by the Ethics Committee. However, it is not mentioned whether these sensors were FDA-approved or not, and no conclusive statement was provided on how rigorously testing was performed by an independent reviewer or institution.

Future work in this field will be focused on developing sensors for continuous monitoring throughout the pregnancy period. The primary focus will be to develop a sensor that can detect any abnormalities in fetal ECG or movement in real-time and give feedback or alert to the patients and the doctor so that the necessary steps can be taken in time and any further complications can be avoided. More flexible sensors, such as the bandage sensors used in the article [7], need to be developed; they will be more convenient and comfortable for pregnant women. There is a need to increase the number of participants for more accurate results. Addressing the limitations and issues of the wearable device will lead to more opportunities to develop wearable sensors, thereby leading the way in commercializing the sensors.

## 6. Conclusions

In this systematic review, we focus on three major research questions, including the sensor’s description and processing, data preprocessing, and performance. The existing sensors were thoroughly examined in this review, and their limitation and future scope were identified. The sensors used in these articles help to monitor maternal health (monitoring MECG, body temperature, and stress level) as well as fetal health (FECG, FM). In the study, we found that pregnant women considered wearing a mobile sensor and reported no privacy concerns. According to Runkle et al., seven in ten women agreed to change their behavior or lifestyle during pregnancy after receiving recommendations or messages from a smartphone through wearable sensors. Most of the studies we mentioned applied Machine Learning and Deep Learning methods to the pre-processed signal to detect or evaluate the performance. The accuracy in these studies was around 80% to 90%, which is acceptable according to the articles. So, we can state that wearable sensors significantly contribute to maternal health during pregnancy. Although there is demand and need for wearable healthcare monitors or sensors, there are still some limitations and challenges. Improving accuracy, testing in free-living conditions, enhancing the comfort level of wearable sensors, and continuous and longitudinal monitoring requires the further development of wearable sensors and devices. If these challenges can be addressed, wearable sensors may substantially contribute to reducing the mortality rate due to pregnancy complications or other maternal health issues.

## Figures and Tables

**Figure 1 sensors-23-02411-f001:**
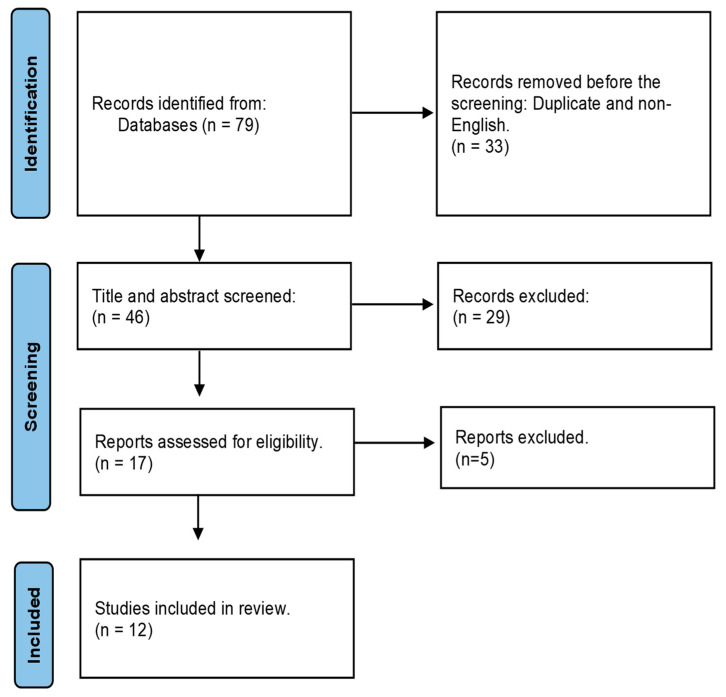
Flow diagram outlining the systematic review strategy.

**Figure 2 sensors-23-02411-f002:**
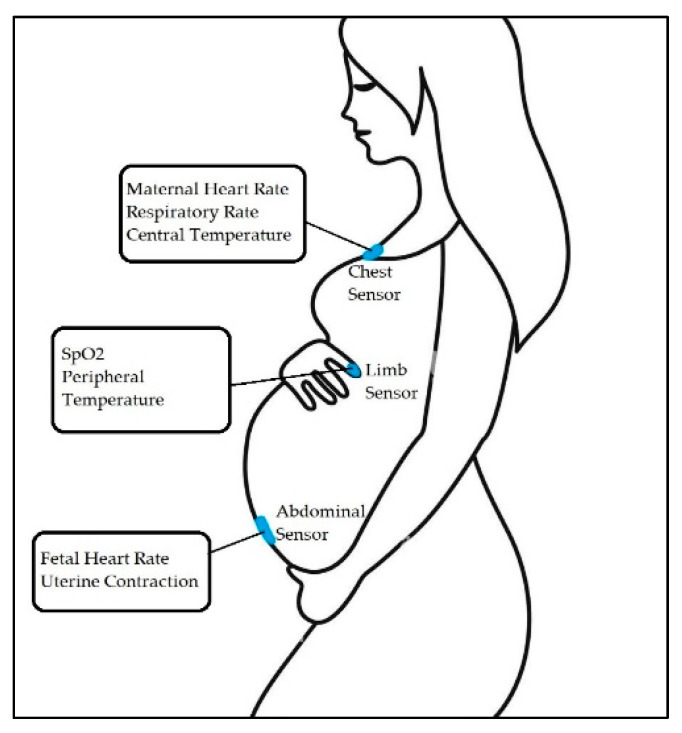
Overview of the maternal-fetal monitoring system, where chest and abdominal sensors capture unique signals from the patient [7].

**Figure 3 sensors-23-02411-f003:**
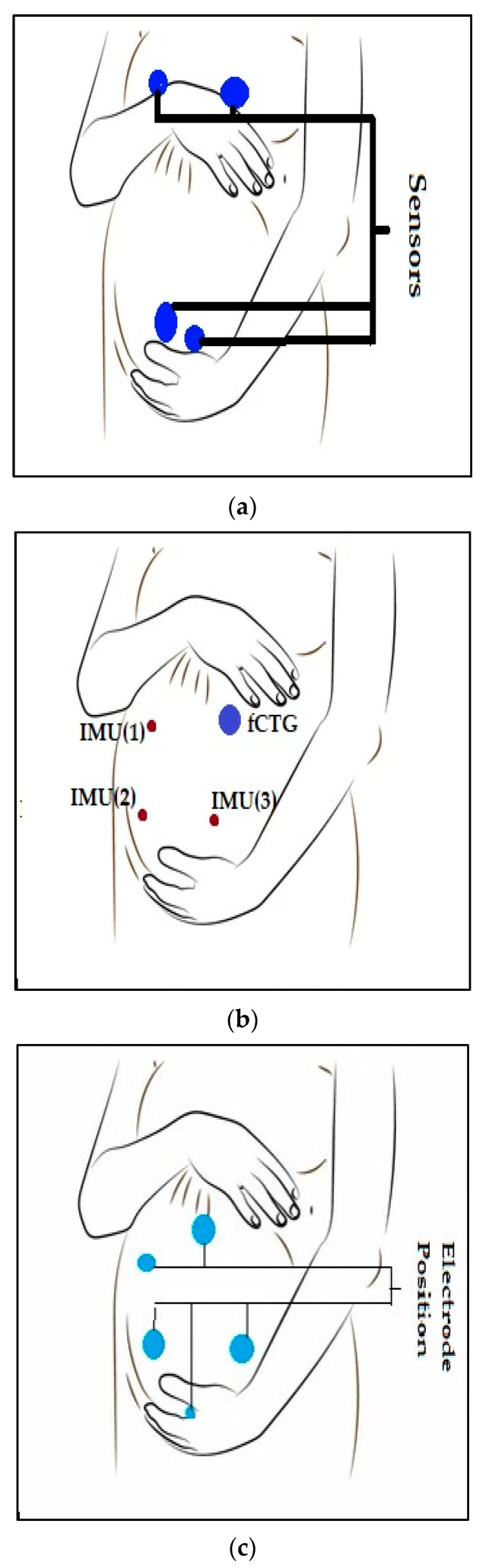
Electrodes or sensor nodes placements for FECG and MECG recording. (**a**) electrodes placement for maternal and fetal ECG recordings for study [20], (**b**) Electrode placement for study [22], (**c**) setup with three sensor nodes for study [24].

**Figure 4 sensors-23-02411-f004:**
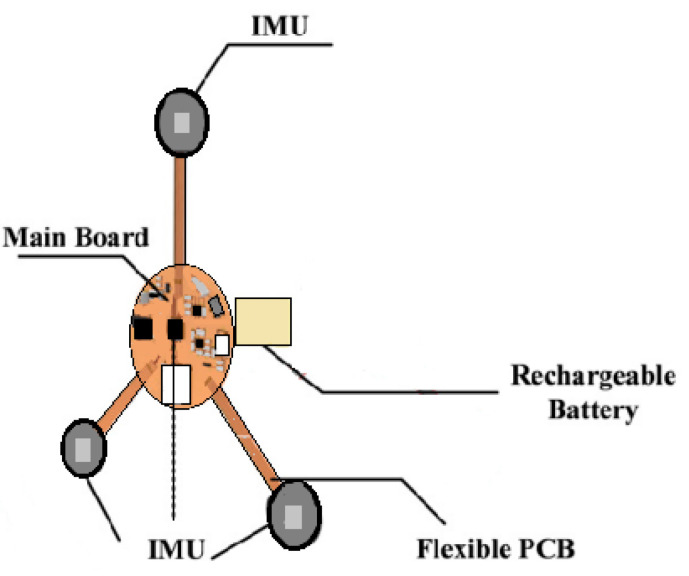
The wearable design proposed in this study [25].

**Figure 5 sensors-23-02411-f005:**
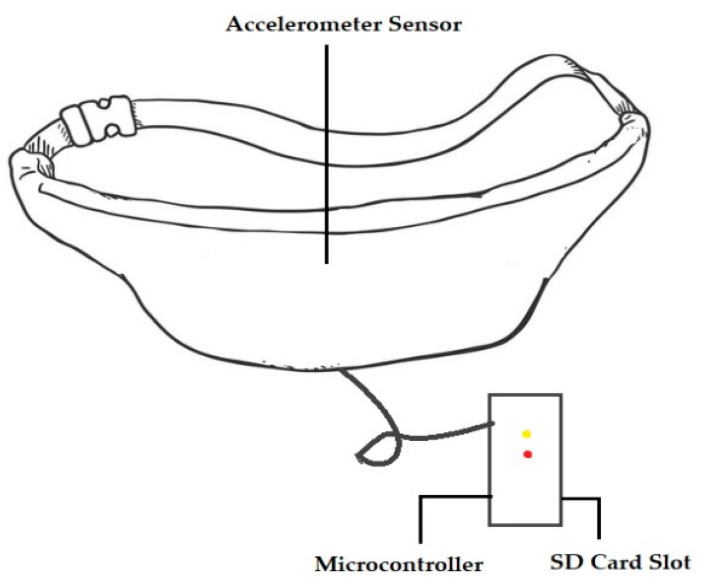
A device used in clinical testing for monitoring fetal movement [27].

**Figure 6 sensors-23-02411-f006:**
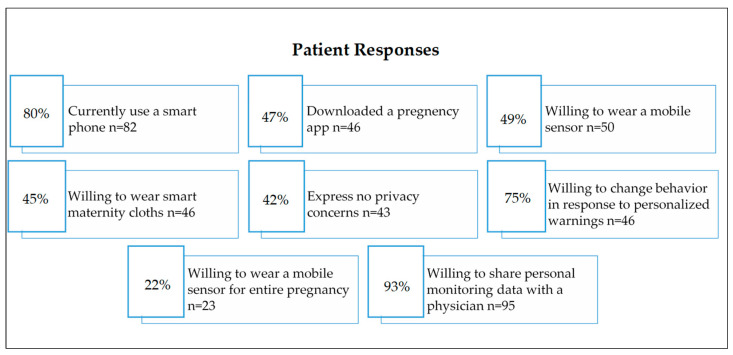
Patient and provider’s response to using digital devices or wearable sensors [3].

**Figure 7 sensors-23-02411-f007:**
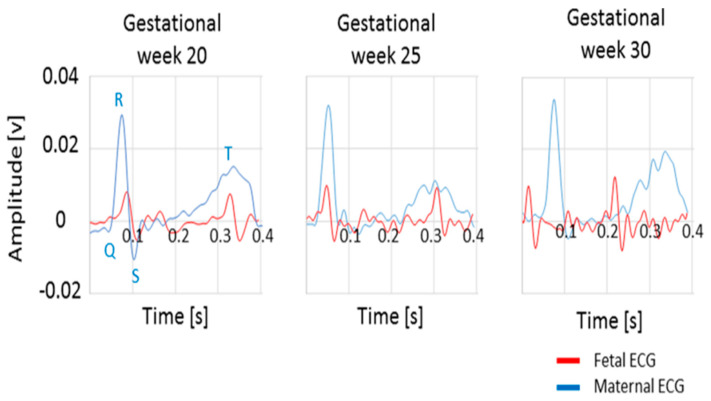
The QRS complex of raw fetal and maternal ECG traces for gestational weeks 20, 25, and 30 [20].

**Figure 8 sensors-23-02411-f008:**
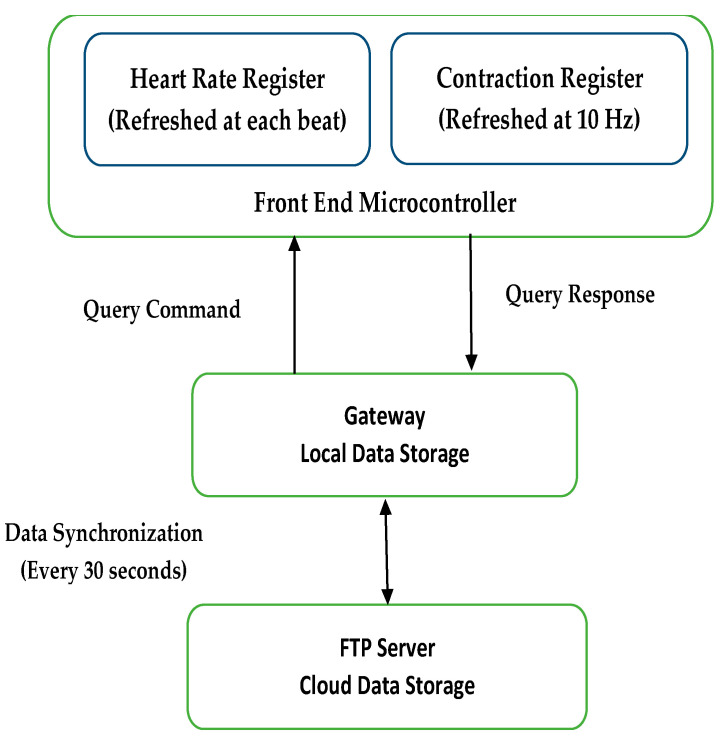
Timing of data transmission in the system [21].

**Figure 9 sensors-23-02411-f009:**
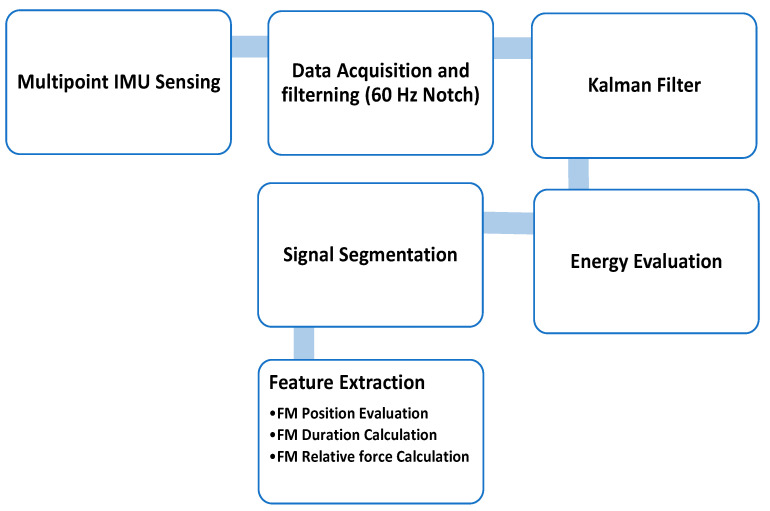
System structure diagram and the features of fetal movement [25].

**Table 1 sensors-23-02411-t001:** Selected articles, their publication date, and total citation.

Ref	Publication Date	Citation Number
**[3]**	2019	41
**[4]**	2021	65
**[7]**	2021	53
**[20]**	2018	30
**[21]**	2011	6
**[22]**	2022	42
**[23]**	2022	33
**[24]**	2021	41
**[25]**	2021	40
**[26]**	2022	32
**[27]**	2020	15
**[28]**	2018	42

**Table 2 sensors-23-02411-t002:** Inclusion and exclusion criteria for this systematic review.

Inclusion Criteria	Exclusion Criteria
Articles published in peer-reviewed venues	Papers not written in English.
Articles published since 2012	Articles not included wearable sensors.
Articles must address a certain combination of words, i.e., (sensor/wearable) + (maternal/pregnant women) + (fetal movement/ECG/HR) + (monitoring/physical activities)	

**Table 3 sensors-23-02411-t003:** Outline of sensing modality and acquired signal.

Ref	Sensing Modality	Acquired signal
**[7,20,21,22,23]**	FECG with/without MECG	Fetal heart rate (FHR)
**[24]**	Seismocardiogram (SCG), Gyrocardiogram (GCG)	Fetal heart rate (FHR)
**[25,26,27]**	Position and Activity	Fetal movement (FM)
**[4]**	Hand Movement	Mother physical activities
**[28]**	Heart Rate Variability (HRV), Heart Rate (HR)	Maternal Stress

**Table 4 sensors-23-02411-t004:** The information on the data collection of the above-mentioned papers.

Ref	Data	Gestational Period	Participants	Experimental Setup
[7]	Maternal ECG, SCG, FECG, EHG	Between 25 and 41 weeks	576	Daily regular activities
[20]	Maternal and fetal ECG	20, 25, and 30 weeks	Not mentioned	Recordings were carried out in a private room. The time period was 30 s and repeated 5 times.
[24]	FCTG, FHR	Not mentioned	10	The subjects were required to stay in the supine position, seated position, and standing position. Five minutes for each position.
[21]	FHR	38.5 weeks	Not mentioned	The system has been tested on both prenatal and laboring patients.
[22]	MECG, FECG, and fetal QRS complex	37 weeks	3	The experiment protocol consists of 3 steps. (1) Supine situation for four minutes, (2) sitting position for two minutes, (3) standing posture for two minutes. The experiment is designed in a home-like laboratory environment.
[23]	FHR	>12 weeks	4	There were two stages. (1) a set of measurements was obtained by employing PPG sensors of light-based technology, (2) results were verified with a professional cardiograph (golden standard)

**Table 5 sensors-23-02411-t005:** The information on the FM data collection of the above-mentioned papers.

Ref	Gestational Period	Participants	Time
[25]	Over 28 weeks	13	1 h
[26]	Over 38 weeks	20	30 min
[27]	28 to 40 weeks	Initially 77	20 min

**Table 6 sensors-23-02411-t006:** Sensors’ description of the above-mentioned papers.

Ref	Sensors	Description
**[7]**	Accelerometer, clinical-grade thermometer, and pulse oximetry module	An accelerometer sensor measures the acceleration of anybody or an object [49]. A thermometer is used to measure and display body temperature [50]. A pulse oximeter is used to monitor the amount of oxygen carried in the body [51].
**[24]**	Commercial IMU	Shimmer 3 from Shimmer Sensing (accelerometers and gyroscopes) [38]
**[20]**	EPS Sensor	It is a feedback-enhanced and stabilized electrometer-based amplifier that operates based on current displacement measurements.
**[21]**	Toco Sensor	Measures the tension of the maternal abdominal wall [52].
**[22]**	AgCl electrode	It has good conductivity, low noise, and a stable baseline.
**[23]**	PPG sensor	An uncomplicated and inexpensive multi-point measurement method that is often used for heart rate monitoring purposes [53].
**[25]**	Multi-point IMU	Used to measure acceleration, angular velocity, and magnetic fields [54].
**[26]**	Two mCube MC3672 accelerometers	Ultra-low-power, low-noise, integrated digital output 3-axis accelerometers with a feature set optimized for wearable [55].
**[27]**	Tri-axial accelerometer	Provides simultaneous measurements in three orthogonal directions [56].
**[4]**	Accelerometer, gyroscope, and temperature	An accelerometer sensor measures the acceleration of anybody or an object. A gyroscope senses angular velocity [49,57].
**[28]**	Garmin vívosmart 2 smart bands	The device monitors heart rate at the wrist and includes helpful tools such as all-day stress tracking [58].

**Table 7 sensors-23-02411-t007:** Description of filters used in the mentioned papers.

Filter and Function	Description
IIR	The infinite impulse response (IIR) filter is a recursive filter in that the output from the filter is computed by using the current and previous inputs and previous outputs.
CWT	Continuous Wavelet Transform (CWT) is a technique for analyzing signals. It uses inner products to measure the similarity between a signal.
Kalman filter	It is an algorithm that provides estimates of some unknown variables given the measurements observed over time.
BlockJS	This method is based on determining an optimal block size and threshold of a signal. More information can be found in [61].
LevelIndependent	This method estimates the variance of the noise based on the finest-scale (highest-resolution) wavelet coefficients [61].
Hampel	The Hampel Filter block detects and removes the outliers of the input signal [62].
Haar, sym2	Valid built-in orthogonal wavelet family for denoising. More information can be found in [61].

**Table 8 sensors-23-02411-t008:** Overview of the analysis.

Ref	Sensors/Device	Monitoring	Participants	Communication	Storage
**[4]**	Accelerometer, gyroscope, and temperature	MPAR	61	Wi-Fi	Raspberry-PI device memory
**[7]**	Chest, Limb, and Abdominal sensors	FHR, MHR, temperature	576	Android or iOS	Mobile Device
**[24]**	Three commercial wearable sensor nodes	FHR	10	MATLAB	Memory Card
**[20]**	EPS Sensor	FECG	-	Laptop	LabVIEW software
**[21]**	Toco Sensor	FHR	-	Bluetooth module, Wi-Fi	Gateway
**[22]**	portable, home-based FECG monitoring device	FECG	3	Bluetooth	Memory Card
**[23]**	Prototype wearable device	FHR	4	Internet	ThingSpeak.com [64]
**[25]**	IMU sensors	FM	13	Wire transmission	SD card inside the device
**[26]**	Two mCube MC3672 accelerometers	FM	20	SoC and smartphone	Laptop/PC
**[27]**	Tri-axial accelerometer	FM	77	Mobile phone	Micro SD card
**[28]**	Garmin vívosmart 2 smart bands	Heart rate at rest (Stress)	20	Internet	Virtual Private Server (VPS)

**Table 9 sensors-23-02411-t009:** Description of the research analysis.

Ref	Pre-Processing	Feature Extraction	Methods	Accuracy
**[24]**	Band-pass filter and IIR pass band filter	CWT (Continuous Wavelet Transform)	Cepstrum method	Reliability from SCG is 75.02% and GCG is 75.52%
**[21]**	Embedded filter on a microcontroller	-	Benchtop tests	The Concordance Correlation Coefficient of FHR is 88%
**[22]**	Notch filter, Butterworth filter, and median filter	QRS Detection	Adaptive Dual Threshold (ADT) and Independent Component Analysis	The average Se, PPV, ACC, and F1 score are 99.62%, 97.90%, 97.40%, and 98.66%, respectively
**[25]**	Notch filter, Kalman filter	FM Duration Calculation, FM Relative Force Calculation, FM Position Evaluation	a phantom test and clinical trial	Accuracy 90.3%
**[26]**	Wavelet, sym2, Denoising Method, BlockJS, NoiseEstimate, LevelIndpendent	Min, max, SD, mean, and median. absolute area, relative area, absolute area of the differential, entropy, and kurtosis	k-fold cross-validation	Accuracy 86.6%
**[27]**	Embedded microcontroller filter	Extracting a laughing or kicking window, Extracting a normal window, Numerical step	Deep Learning: STFT combined with CNN	Accuracy 73% and 88%
**[28]**	Digital filtering		K-means clustering, Random Forests	Accuracy 97.9%
**[4]**	-	Mean, SD, cosine similarity, RMS, Skewness, Kurtosis, Max value, Min value, Frequency Domain Features, Entropy, Zero Crossing, Quartile Range, and Absolute Time Difference between Peaks.	K-Nearest Neighbors, Decision Tree, Random Forest, Induction rules, and Gradient boosted trees.	Accuracy 89%

**Table 10 sensors-23-02411-t010:** Pros and Cons of the sensor modalities used in the articles.

Ref	Pros	Cons
**[4]**	Adjustable like a wristwatch and has low battery power consumption.	Sensors positioned at the wrist position, that were used with only ten activities, did not consider other parameters such as ECG and FM.
**[7]**	Sensors are low-cost, flexible, comfortable, and suitable for real-time measurement.	Did not include FM in the study.
**[20]**	It is non-invasive in nature; dry electrodes that do not require skin preparation or gels to be applied on the body and reduce moving artifacts.	The recording time was very short.
**[21]**	It has a mobile cellular gateway for a wide range. Communication and browser-based user interface for remote monitoring and diagnostics and a low cost	The sensors are designed with straps, adjustable belts attached to external Bluetooth, and a central unit that can be a little uncomfortable for pregnant women. The number of participants was small.
**[22]**	Consists of biocompatible electrode materials, noise suppression design, an amplification circuit, data transmission, and a storage module.	Tested for short-time recording; the number of subjects was small, and power analysis conducted was not performed.
**[23]**	It is a cheap and reliable system; the user does not need to trigger an alert manually, making it easy to operate for the end user and the clinician.	The was in a preliminary stage and the system needs to be widely tested; the development board size and weight needs to be further reduced
**[24]**	Three different measurement positions were evaluated and compared for each result.	The test was conducted at rest and was not suitable during movement for motion artifacts.
**[25]**	The design configuration is flexible.	The participant had to click the counter when they felt fetus movement.
**[26]**	This algorithm could handle motion artifacts, proving its robustness	The number of participants was small
**[27]**	Low-cost device; the sensors are non-invasive and non-transmitting. As such, it can be used over an extended period with no negative health impact.	The data were only recorded for 20 min.
**[28]**	This study was able to implement a real-time stress level estimation algorithm, and it worked in an online setting.	Most of our data from this study do not have true labels.

## Data Availability

Not applicable.

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
