# Peer review of "Wearable Sensors for the Monitoring of Maternal Health—A Systematic Review"

_sensors, 2023, doi:10.3390/s23052411_

Round 1
Reviewer 1 Report
In this review, the focus is on a systematic evaluation of wearable sensors for monitoring maternal health in daily life. The goal is to describe the usefulness and limitations of current methodologies, as well as identify areas for further research. This interesting topic comes within the scope of the Sensors Journal.
The paper, in this reviewer's opinion, needs to be improved:
1) After reading this paper, I believe the writers should expand on the original sources, such as Science Direct and IoP Science. The topic is quite interesting, but the quantity of publications analyzed is relatively low, which may lead to the neglect of other sensing technologies or approaches.
2) How many papers were reviewed in total?
3) The following systematic reviews should be implemented: 1) Cochrane Handbook for Systematic Reviews of Interventions, http://handbook-5-1.cochrane.org/, 2) Drinking frequency effects on the performance of cattle: a systematic review, DOI: 10.1111/jpn.12640.
Reviewer 2 Report
An article with interesting content. The authors showed knowledge of the subject matter and familiarized themselves with the works of other researchers.
The article refers to the analysis of selected research works in the field of monitoring maternal health taking into account the characteristics of wearable sensors and devices. However, I have a few comments. Please refer to them.
1. There is some dissatisfaction with the planned review of works and the methodology of the review. The "systematic review of papers" mentioned in the paper, according to good practice, should include methodology: identification, analysis, and interpretation of all available data. The strategy for the selection and classification of papers could be more broadly described, i.e., presenting the results obtained in stages. Currently, the paper refers only to the final result of the search and article classification criteria. Are similar reviews from this area known? What approaches have been used by other researchers? Has the SCIMago index or citation index been tracked?
2. There are numerous abbreviations in the paper in many places. My impression is that not all of them have been explained, which will lead to unnecessary conjecture. Abbreviations should be explained where they are first used or in the index of abbreviations.
3. The paper cites papers with a discussion for some of them of many different types of filters and data processing algorithms. For many of them, it would be good to cite the availability of the data used or collected by the authors of the papers discussed, in order to perhaps deepen this analysis with their own research.
4. The discussion mostly presents a summary of what is written in the article in the form of dry facts and plans for the future. What is sorely lacking there is the author's evaluation of the works read and a word of commentary comparing them with each other. After reading such a discussion, the reader should know which of the presented works is valuable.
5. The design challenges of wearable sensors in maternal health monitoring should be listed as the author's opinion with regard to the review conducted.
6. Conclusion does not contain conclusions, but only the motivation for writing the paper.
Reviewer 3 Report
This review provides an overview of the important role of wearable sensors for maternal health, as well as a summary of effective monitoring methods and the principles of wearable sensors. The summary and research in this area are interesting for the reader. However, the review requires major revision before being considered for publication. The quality of the manuscript writing should be further improved, and a more in-depth discussion is expected.
1. The figure arrangement requires major revision to fulfill requirement for a review. 1) It is suggested to design a figure to indicate the overview of one review in Figure 1.
2) It is a normal practice to organize sub-figures that deliver the similar key message in one figure. For example, figure 2-4 that discuss electrode positions can be incorporate into one figure. Same for the other separated figures in the review. 3) Fig. 1C uses the same image as Fig. 9B
4) The scales of figures are not properly arranged. For example, Fig. 7, Fig. 10 and Fig. 11 are not proportional scaled in x and y axis.
2. The tables are too simplified to be declared as ‘summary’. It is not suggested to include long sentences in tables. Besides, when summarizing and comparing difference measure techniques or strategies, key factors/benchmark values/records like sensitivity, advantages, reliability should be included.
3. Section 5 describes that these sensors are safe for both mother and baby. It is suggested to include more supportive data and discussion.
4. The conclusion section is too simplified. A brief summary on this review, challenges and prospects for advances in this field should be included.
5. Please ensure that the figures in this manuscript are of high resolution as requested by the journal.
Reviewer 4 Report
Authors are appreciated to take this study related to Wearable devices for maternal health monitoring.
The following are the observations and suggestions to improve the quality of the article.
In section 2.2 IEEE Explore has to be mentioned as IEEE Xplore
Table 3: The summary of the data collection of above mentioned papers
Table 4: The summary of the FM data collection of above mentioned papers
The references included related to the filters are very generic not related to this study , not explained related to the study.
Grammatic and spelling corrections has to be carried out.
The interpretation and scope for research /improvement are very superficial has to be well highlighted as this is the literature study to kindle the interest of the reader/beginner related to this domain should get clarity and get the starting point for their research.
The article has to carefully read and appropriate points has to highlighted technically with research scope. The detailed study on pros and cons of the SOTA methods has to be presented to attract more readers for this article.
Suggested to include the research literature articles related to signal conditioning and monitoring related to this .
Round 2
Reviewer 1 Report
There are no more observations.
Author Response
We would like to thank the reviewer for the useful comments and suggestions
Reviewer 3 Report
The figures arrangement is not in a professional way as for review.
